# Developmental and Housekeeping Genes: Two Types of Genetic Organization in the *Drosophila* Genome

**DOI:** 10.3390/ijms25074068

**Published:** 2024-04-06

**Authors:** Igor Zhimulev, Tatyana Vatolina, Victor Levitsky, Anton Tsukanov

**Affiliations:** 1Institute of Molecular and Cellular Biology of the Siberian Branch of the Russian Academy of Science, 630090 Novosibirsk, Russia; vatolina@mcb.nsc.ru; 2Institute of Cytology and Genetics of the Siberian Branch of the Russian Academy of Science, 630090 Novosibirsk, Russia; levitsky@bionet.nsc.ru (V.L.); tsukanov@bionet.nsc.ru (A.T.)

**Keywords:** polytene chromosomes, housekeeping genes, developmental genes, promoter motifs, *Drosophila*, bands, interbands, CHRIZ/CHROMATOR, transcription factor binding site, de novo motif search

## Abstract

We developed a procedure for locating genes on *Drosophila melanogaster* polytene chromosomes and described three types of chromosome structures (gray bands, black bands, and interbands), which differed markedly in morphological and genetic properties. This was reached through the use of our original methods of molecular and genetic analysis, electron microscopy, and bioinformatics data processing. Analysis of the genome-wide distribution of these properties led us to a bioinformatics model of the *Drosophila* genome organization, in which the genome was divided into two groups of genes. One was constituted by 65, in which the genome was divided into two groups, 62 genes that are expressed in most cell types during life cycle and perform basic cellular functions (the so-called “housekeeping genes”). The other one was made up of 3162 genes that are expressed only at particular stages of development (“developmental genes”). These two groups of genes are so different that we may state that the genome has two types of genetic organization. Different are the timings of their expression, chromatin packaging levels, the composition of activating and deactivating proteins, the sizes of these genes, the lengths of their introns, the organization of the promoter regions of the genes, the locations of origin recognition complexes (ORCs), and DNA replication timings.

## 1. Introduction

*Drosophila* polytene chromosomes are permanently in interphase, and genes are actively expressed throughout this stage. They have a transversal pattern of bands and interbands, as well as giant size. The first drawing of *Drosophila melanogaster* polytene chromosomes was made under a light microscope by T. Painter in 1934 [1]. While non-polytene interphase chromosomes have two DNA strands each, the polytene ones consist of thousands of identical DNA strands arranged into bundles, and for this reason are giant. A polytene chromosome has a transverse banding pattern, which is due to gray and black bands of different lengths. The differences in shades represent the differences in the packing density of the DNA in them. Darker bands are interspersed by lighter stripes (or interbands). Bands and interbands have different lengths and thicknesses as well as shades of gray (mostly black and gray), which grant each region its unique morphological presentation, allowing the researcher to locate a band of interest or a gene of interest in it, create a map of bands, and assign names to them (Figure 1).

Although black and gray bands are so well distinguishable that they were described as early as first publications were made (the authors benefited from electron microscopy) (Figure 1), these distinctions in color and morphology have long been overlooked.

Any genome is made up of at least two types of genes: those being expressed permanently, to control basic cellular functions (housekeeping genes or ubiquitously active genes), and those that are active only in certain tissues or at certain stages of development (developmental genes). However, little is known in detail about how these groups of genes are organized. The main problem with the situation is that they are difficult to identify. Among the first attempts to assign these types of genes to some chromosome structures, two works from the early 1970s deserve particular mention [4,5]. At that time, it came to the authors’ attention that for the genes to be permanently active, the chromosome material should be permanently open (or decondensed), i.e., transcription-ready. This property is possessed only by interbands (Figure 2), while the genes activated by the hormone ecdysone before molting lie in the bands that will become puffed out.

According to C. Speiser [4], there was a “basic genome” that came first in evolution, and later its genes spawned developmental genes, which are now seen as black bands on the polytene chromosomes. To go eukaryotic, living things should have started with a “primordial cell”, which was supposed to possess a “primordial genome” made up of housekeeping genes (see also Figure 2). Thus, in these two works [4,5], all genes of the genome were hypothetically divided into two groups: developmental and housekeeping genes, and possible chromosome structures, where they could be localized were proposed: in the interbands of housekeeping genes and in bands of developmental genes. As the material of this review is further presented, conclusions will be substantiated first about the actual location of two groups of genes in chromosome structures, then about the creation of a mathematical model that allows us to identify, select, and perform high-accuracy mapping of genes belonging to both groups on chromosomal and physical maps, and finally study the structural features of the genes included in each group and identify differences in their structures.

## 2. The 4HMM Model of Chromatin States in Chromosomes and Assignment of Genes to Chromosome Structures

In order to study the genetic organization of chromosome structures (two types of bands and interbands) and gene groups located in these structures, it is important to know the exact localization of the boundaries of chromosome bands and interbands on the molecular (physical) map of DNA. Then, it is possible to determine which genes are located in a particular structure within its boundaries. This has not been performed yet. Conducted mainly in the 1970–1980s, experiments on saturation of certain chromosome regions with mutations, usually within small deletions, followed by comparison of the identified number of genes with the number of bands and interbands in the region, gave the well-known direction “one gene—one band” (see [6] (pp. 94–148) for details, discussion, and reference). However, correlations between the number of genes and the number of bands say nothing about the genetic organization of bands or interbands, since in each of the studied regions, there is the same number of both bands and interbands, as well as band–interband complexes; as a result, it is not clear where exactly the genes are located (see [6] (p. 114) for more details).

Although nucleotide sequences of the *Drosophila* genome had been determined by 2000 [7], many releases have been published since then (for some of the latest and most complete, see [8,9]). This knowledge also did not lead to identification of the boundaries of chromosomal structures on the physical map, since the sequence gave a long sequence of nucleotides and the position of genes on it, but nowhere on it are there any areas specific for the transition from bands to interbands in the chromosome, so the position of bands and interbands remained uncertain. After the discovery of mobile elements in the *Drosophila* genome [10,11] and creation of artificial (or experimental) transposons (DNA fragments containing various sequences in various vectors) (the P-lArB, transposon containing the terminal sequences of the P-element, and parts of the *lac* operon of *E. coli* and parts of the *Drosophila* genes *ADH and rosy* [12] can be provided as an example), it became possible to analyze the sites of transposon insertions using electron microscopy (EM). 

It was established that transposon DNA in *Drosophila* cells formed a new band ([6] (p. 91), [13]) if the insertion occurred between the native polytene chromosome bands (Figure 3A,C). If a transposon is inserted into a polytene chromosome band, the condensed material of the transposon and the native band is combined, and a new band does not appear (Figure 3B,D). By sequencing DNA in the site of transposon insertion in the interband, it is easy to locate its insert on the DNA map with high accuracy; and at the same time, we know that this is the interband DNA. Therefore, transposon insertion in the case when a new band is formed is a molecular hallmark of an interband on the physical map of DNA. Fifteen *Drosophila* lines containing insertions of various transposons into interbands (P-lArB, etc.) were taken for the study [2,6,13,14]. Another six interbands were mapped using EM in the region 9F-10B, six more in regions 7F, 19E, 35D, 56A, 58A, 70A, and another five (three in the region 21D and 100B) were mapped using EM and FISH [2,6,13,14]. A single deletion in *Drosophila* was also used, precisely mapped at the light microscope level as removing only one interband, the *facet^strawberry^* [15,16]. The coordinates of all the 33 studied interbands on the physical map of the *Drosophila* genome are indicated in [2].

Later, the positions of many interbands in polytene chromosomes were confirmed using the Hi-C method for localizing interTAD and TAD regions [17,18,19,20]. 

To identify the position of proteins, histone modifications, proteins of origin recognition complexes of replication (ORC), and genetic elements in the interband sites and to create possible groupings of them, we used the data obtained during the implementation of the modENCODE program (model organism ENCyclopedia of DNA Elements) [21]). We also used data from three *Drosophila* full genome chromatin classifications which have been developed, with individual chromatin types represented by a unique combination of proteins, their functional specificity, as well as characteristic genomic position. They are briefly described below:

(*i*) Using DamID analysis of 53 chromatin proteins, 5 major chromatin states were established [22]. In this model, transcriptionally active chromatin is represented by the RED and YELLOW types. Notably, YELLOW chromatin is enriched with housekeeping genes and histone mark me3. This histone modification is associated with transcription elongation and localizes in gene bodies, predominantly in exons [23,24,25].

(*ii*) The distribution of 18 histone modifications in the chromatin of 2 *Drosophila* male cell lines has uncovered 9 chromatin states [26]. Active chromatin here is represented by states 1 and 2 (red and purple, respectively), and in the context of the X chromosome, by state 5 (green) that marks the regions associated with dosage compensation machinery. Similar models were subsequently developed for the human [27,28] and *Arabidopsis* genomes [29]. 

(*iii*) One more chromatin classification model is based on the differential sensitivity of chromatin to DNase I and broadly classifies chromatin into open, neutral, and closed states [30].

(*iv*). Our classification of chromatin was based on the principle of the presence of proteins, histone modifications, and genome elements in chromosome structures depending on the degree of chromatin condensation in them. We already knew that the degree of chromatin folding was minimal (Figure 4) in the interband regions while being maximal in the black bands [31,32].

First, we selected proteins and histone modifications that were accurately mapped in active (open) chromatin and, at the same time, mapped in the interbands of polytene chromosomes using antibodies. These were histone modification H3K36Me [33,34], RNA polII [35,36], Myc [37], insulator proteins BEAF-32 [38], Mod (mdg4), Su(Hw), Zw5, CTCF, CP190 [39,40,41], promoter-associated proteins Z4 and Chriz [42], and proteins of Origin Recognition Complex (ORC2 and ORC6) [43,44,45]. These and other active chromatin proteins (see above) were placed on the hit map [2], and we focused on the distribution and analysis of chromatin types that were identified using 4HMM (hidden Markov model). This model takes into account binding data for the “open chromatin” proteins and uses the data obtained for Kc, S2, BG3, and Clone8 cell lines as an input. As a result, four basic chromatin types referred to as cyan, blue, magenta, and green have been identified [2]. In order to avoid possible confusion with color-coded chromatin types published by other groups, we had our chromatin types renamed as follows: aquamarine (formerly, cyan), lazurite (formerly, blue), malachite (formerly, green), and ruby (formerly, magenta) [46].

According to this model and data from subsequent studies, the composition of aquamarine chromatin, localized in the interbands of polytene chromosomes, has the most extensive list of proteins and histone modifications (Figure 5).

The most abundant chromatin state in the genome is ruby (black bands) (48%), and the least abundant is aquamarine (interbands) (13%) (Figure 6A).

Inspection of these chromatin types clearly indicated that aquamarine was related to interbands, with notable enrichment for the interband-specific protein CHRO (CHRIZ) ([2] and references therein) and 5′-ends of genes; lazurite chromatin matched gene bodies and morphologically corresponded to gray bands flanking the interbands (Figure 7).

A conclusion can be drawn that a housekeeping gene occupies two polytene chromosome structures: the promoter resides in the interband [2,47,48], and the gene body lies in the gray band downstream of it. These data were supported by our whole-genome analysis of chromatin state distribution. Aquamarine fragments were found to always occur next to lazurite [2] and always reside upstream of it (Figure 7) [47,48]. MSL1 protein is located in the gene bodies during transcription; lazurite chromatin contains transcription elongation factors, and electron reveals gold-conjugated antibodies against the MSL1 protein in gray bands, that is, in Lazurite chromatin [49].

The ruby chromatin corresponded to the repressed material of black bands [2]; malachite chromatin was found on the edges of ruby chromatin fragments, intergenic spacers, and in introns [46].

## 3. Housekeeping and Developmental Genes

Transcriptional activity in DNA fragments that are a part of 33 interbands (aquamarine) was studied by analyzing the databases for ChIP-seq data (see [2,47], and references therein). Proteins in chromosomes were identified using antibodies, and the location of the read-out gene was determined by nucleotide sequencing. By counting the transcripts in different *Drosophila* cell types specific for a particular developmental stage, tissue or organs of larvae and adult flies, cell cultures, and the like, it was shown that transcriptionally active genes are largely located in interbands (aquamarine chromatin) (Figure 8A), while very few of them reside in the 33 black bands arbitrarily chosen as the control group (ruby chromatin) (Figure 8C). The number of transcripts in aquamarine chromatin is 22–27 times higher than that in ruby chromatin. A similar analysis on aquamarine and ruby fragments in the whole genome of *Drosophila* yielded the same results (Figure 8C,D). Thus, the genes residing in visible interbands and in all the aquamarine fragments genome-wide are ubiquitously active and can therefore be regarded as candidate housekeeping genes. On the other hand, the transcripts produced from black band genes (those residing in ruby chromatin) were scarce in any tissue, suggesting that these genes are active only locally and are most likely to be developmental genes.

For gaining a deeper understanding of the biological functions of protein-coding genes that localize to four different chromatin states across the *Drosophila* genome, we used DAVID bioinformatics resources [52] to analyze large gene lists. The basic analysis strategy is to systematically match a large number of genes in a list with the corresponding biological annotation and then statistically isolate the most represented (enriched) biological annotation out of thousands of linked terms and contents. Enrichment analysis is a promising strategy increasing the chances that researchers will be able to identify biological processes that are most relevant to the gene lists being studied.

GO term enrichments were calculated for biological processes. We first identified the fraction of genes belonging to the GO term for lists of genes respecting four chromatin states. The top representative terms are shown in the histograms (Figure 9). One can see that the genes whose promoters are associated with aquamarine chromatin are enriched with significant terms for biological processes compared to all other genes in the *Drosophila* genome (rate (FDR) < 0.05 (Benjamini–Hochberg correction, a built-in function of the DAVID tool).

Although the size of the gene lists is comparable in this analysis (6562 aquamarine genes and 5664 ruby, malachite, and lazurite genes), genes with promoters in aquamarine chromatin are significantly enriched for significant terms compared to the rest of the *Drosophila* genes (Figure 9).

As it turned out, genes with promoters in aquamarine chromatin are enriched with functions associated with biological processes of metabolism of protein molecules, cell organelles, RNA processing and transport, cell cycle and cell division, transcription and expression of genes, apoptosis, replication, amplification and DNA repair, cellular response to stress, chromatin organization, and signaling.

Genes with promoters in ruby, malachite, and lazurite chromatins are associated with metabolic processes such as proteolysis, oxidation–reduction process, flavonoid biosynthetic process, lipid catabolic process, and peptide catabolic process. These genes are also involved in perception of smell, taste, chemical stimulus, ion transport, cuticle development and chitin metabolic process, multicellular organism reproduction, response to bacterium, behavior, neuropeptide signaling pathway, response to carbon dioxide, circadian rhythm, etc. (Figure 9). The housekeeping and developmental genes in *Drosophila* are characterized using a set of enhancers and transcriptional cofactors which display specificity for distinct types of core promoters. It was shown that transcriptional cofactors display specificity for distinct types of core promoters. Enhancer-core-promoter specificity separates developmental and housekeeping gene regulation. Developmental and housekeeping transcriptional programs are different ([53,54,55,56,57,58] and references therein).

Using gene expression data from humans and ten other animal species, including primates, chickens, and *C. elegans* (15 transcriptomes from 11 organisms), they found that stably expressed genes are not necessarily essential (needed for survival) and that individual genes that are essential and are stably expressed can vary significantly between organisms; however, pathways associated with these genes are conserved. In addition, the level of conservation of housekeeping genes in the analyzed organisms reflects their taxonomic groups, demonstrating evolutionary relevance for our definition [59].

## 4. The Organization of Two Gene Types in the *Drosophila* Genome

It would be most interesting to compare the promoters of the two types of genes. A study of the promoter regions of the sequences flanking the TSS genes (in different experiments, approximately from 2000 to 4000 genes) was previously carried out using 5′ expressed sequence tags (ESTs) from cap-trapped cDNA libraries to the *Drosophila* genome. The following motifs have been described: Motif 1 (in subsequent works, M1BP), Motif 2 (DRE or BEAF-32), Motif 7 (ZIPIC), Motif 5 (CRP), and Motif 3 (TATA). However, when selecting genes for analysis, the authors did not distinguish between developmental and housekeeping genes [60,61,62].

We have looked into 13,574 protein-coding *Drosophila* genes in FlyBase (R 5.57) and selected only those genes whose 5′-regions (from −300 to +200 bp relative to the transcription start site) contained aquamarine-only material. Using the STREME tool [63], we have now revealed 6562 aquamarine, 874 lazurite, 1628 malachite, and 3162 ruby genes. With STREME, we performed a de novo search for motifs in each of the chromatin states in three tests. We applied the Bonferroni correction to the fragments enriched for motifs: p/N was less than 0.05, where p is the statistical significance of a motif and N is the number of motifs found with STREME [63]. We compared the “enrichment” motifs against the motifs for known transcription factors in the CISBP (http://cisbp.ccbr.utoronto.ca/ (accessed on 1 September 2022)) and JASPAR (https://jaspar.uio.no/ (https://jaspar.elixir.no/) (accessed on 1 September 2022)) databases using the Tom-tom comparison tool (https://meme-suite.org/meme/tools/tomtom (accessed on 1 September 2022)) and found four motifs characteristic of the housekeeping genes and one characteristic of the developmental genes (Figure 9). 

It has been demonstrated that the genes whose promoters lie in polytene chromosome interbands are enriched for functions associated with cell maintenance, while the other genes (making up about half of the *Drosophila* genome) are associated with highly specialized processes unfolding during development [64]. The promoter regions of the housekeeping genes were found to have four specific motifs that may occur in different genes as singletons or in combinations. A considerable portion of interband promoters was devoid of any of these motifs. An analysis performed with gene ontology showed that some groups of interband genes, whose promoters contain a single motif or combinations of motifs, typically have certain functions to perform. The promoter regions of the housekeeping genes were found to contain four specific motifs, while the developmental genes, only one, a TATA box. Type II promoters (TATA-depleted with fixed Motif 1) in the genes with “ubiquitous” expression in Metazoan were described in reviews [65,66]. About 50% of the developmental genes carry neither TATA nor any of the aforementioned motifs in them. About 30% of the housekeeping genes carry none of the aforementioned motifs in them. This suggests the need to keep searching for new approaches and using new databases to analyze promoters of these groups of genes (Figure 10 and details in [64]).

**Figure 10 ijms-25-04068-f010:**
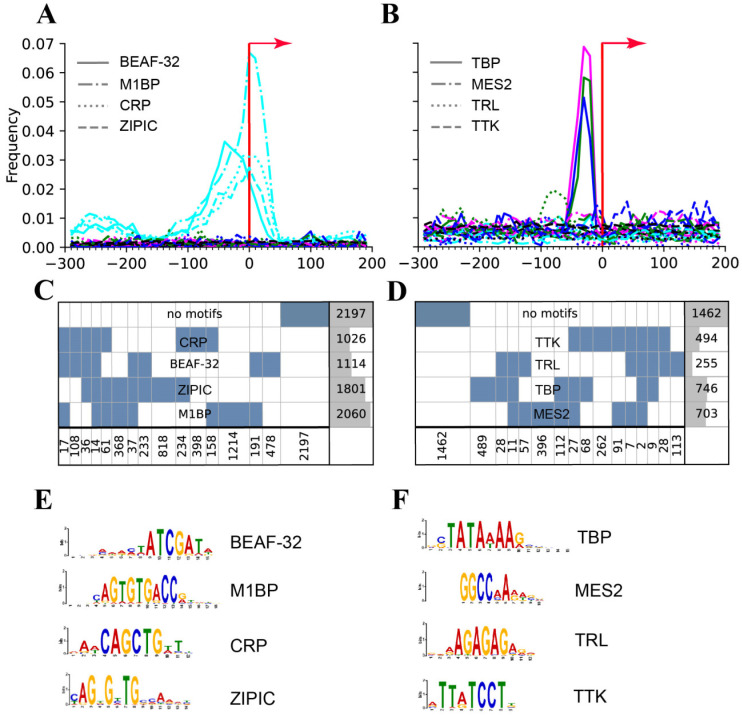
Characteristics of the promoter regions of housekeeping (**A**) and developmental (**B**) genes in the whole genome of *Drosophila*. Positions of the motifs relative the transcription start site (red arrow). *X* axis: the distance from the transcription start site. *Y* axis: the frequency of occurrence of a motif at the given coordinate of the 5′-region. The most used names of the motifs are given in (**A**,**B**) according to CISBP (http://cisbp.ccbr.utoronto.ca/ (accessed on 1 September 2022)) and JASPAR (https://jaspar.uio.no/ (accessed on 1 September 2022)) databases. The motifs occur only in the promoter regions of the “aquamarine” genes (light blue line in (**A**)), the others’ promoter regions not containing these motifs (green, dark blue and ruby lines close to zero on the *Y* axis). The motifs were found in the gene promoters of the rest of the chromatin states (ruby, malachite, and lazurite). Only TBP demonstrates a clear peak in these three chromatin states (**B**). The number of housekeeping (**C**) and developmental (**D**) genes with different sets of motifs in promoter zones (the rightmost columns and the bottom row of numerals) (according to [64]). Sequence logos for binding motifs of proteins BEAF-32, M1BP, CRP, and ZIPIC of housekeeping genes (**E**) and motifs for proteins TBP, MES2, TRL, and TTK of development genes (**F**) found in [64] are referenced in [67,68,69,70,71,72,73,74].

For the housekeeping and developmental genes, we calculated their lengths, the lengths of their introns, exons, and intergenic spacers (Figure 11A), as well as the number of transcripts produced from these genes and the number of exons in these genes (Figure 11B).

The genes in both groups are approximately of the same size; however, the housekeeping genes and their exons are 1.2-fold longer than the developmental genes and their exons. In turn, developmental genes have much longer introns than those in housekeeping genes (4.5 times as long) (Figure 11).

We found that the housekeeping genes are distributed significantly more densely in the genome (*p*-value < 0.05): the intergenic spacers are five times shorter in the housekeeping than in the developmental gene clusters; their median value is 3769.5 bp in the developmental genes and only 706 bp in the housekeeping genes (Figure 11A). Genes of both types are almost of the same size (Table 1). The housekeeping genes are distributed more densely in the genome: the intergenic intervals are five times as long in the developmental gene clusters than in the housekeeping gene clusters; the median length of an intergenic interval is 3769.5 bp for developmental genes and 706 bp for housekeeping genes (Figure 11).

Length distributions also differ significantly between these two types of genes: the median length is 2481 for the housekeeping genes and 2078.5 for the developmental genes. Therefore, the housekeeping genes are about 1.2-fold longer than the developmental ones. This could be because the exons are longer in the housekeeping genes than in the developmental ones: the median exon length is 217 bp in the former genes and 267 in the latter genes (Figure 11A). The introns are significantly (*p*-value < 0.05) shorter in the housekeeping genes than in the developmental ones, the respective median lengths being 78 bp and 309 bp, there being an about four-fold difference. At first glance, it seems reasonable to assume that the housekeeping genes are shorter than the developmental ones; however, this assumption defies reality.

Additionally, we calculated the number of transcripts produced from the housekeeping and developmental genes (Figure 11B). The housekeeping genes produced significantly more transcripts than the developmental genes did, the respective medians being 2 and 1 (Figure 11B).

The transcription initiation system includes many stages, which can be distilled to the following: chromatin decondensation under the influence of insulator proteins [65], assembly of a transcription factor complex on the promoter, transcription per se, with RNA polymerase II and various histone modifications (see reviews [65,66]).

The insulators discovered in *Drosophila* [75,76,77] are able to activate target genes ([77,78,79], (review, [80])). The insulator proteins form associations that bind DNA in varying sites depending on composition of the association of proteins, and CHRIZ/CHROMATOR is a member of this association [81,82,83,84].

CHRIZ/CHROMATOR protein is always located in interbands, i.e., in the transcription start sites of housekeeping genes [2]. Once artificially transferred to the dense black band, it induces decondensation of the band material, which results in the band splitting (Figure 12).

**Figure 12 ijms-25-04068-f012:**
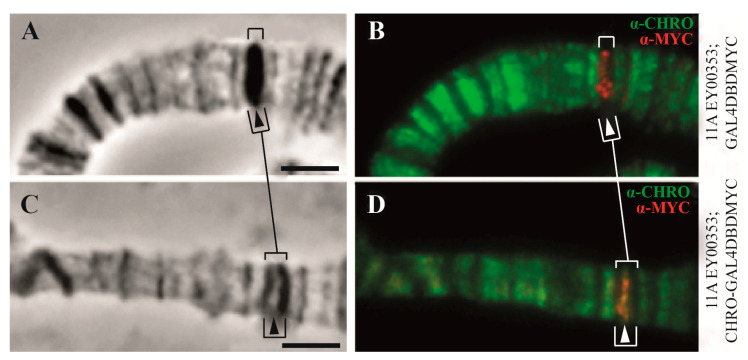
The 11A6-9 band splits upon tethering of CHRIZ/CHROMATOR GAL4^DBD^ (according to [85]). Phase contrast (**A**,**C**), overlay of phase contrast and immunostaining (**B**,**D**). Arrows indicate the 11A6-9 bands in control, where the Gal4^DBDMYC^ protein alone does not split the band (**A**,**B**), although the construct inserted in the band does (red signal of MYC in (**B**). When the CHRIZ/CHROMATOR protein appears in the site of transposon insertion in the 11A6-9 band, it decondenses material of the band in site of insertion, with band splitting and interband formation (arrows in (**C**,**D**)). See [85] for details. Scale in “A” and “C” corresponds to 2 µm.

The CHRIZ/CHROMATOR protein is not observed in black bands when the developmental genes in them are activated and puffing takes place [42]. In the polytene chromosomes, the histone modification H3K36me3, which controls the elongation of housekeeping genes [23,24,25], is found in interbands and gray bands. Antibodies against H3K36me3 and CHRIZ/CHROMATOR co-localize. This modification has not been found in the puffs where developmental genes reside.

The obtained facts indicate large differences in the structure of genes and chromatin in which they are located on chromosomes and associated chromatin states (Table 1).

**Table 1 ijms-25-04068-t001:** Comparison of properties of the developmental and housekeeping genes in the *Drosophila melanogaster* polytene chromosomes.

Genomic and Functional Properties	Type of Genes		References
	Housekeeping Genes	Developmental Genes	
1. Assignment of genes to chromosomal structures	Promoters lie in the interbands (aquamarine chromatin), while gene bodies reside in the adjacent gray bands (lazurite chromatin)	Groups of genes cluster together in black bands (ruby chromatin)	[2,32,47]
2. Packing density of chromosomal material	5–10-fold in promoters (interbands) and 35–70-fold in gene bodies (gray bands)	150–200-fold in developmental genes (black bands)	[31,32]
Openness of chromatin to DNase I	Maximum where housekeeping gene promoter is located	Minimum where developmental genes are located	[30]
Chromatin composition	Proteins and histone modification characteristics for open chromatin	Composition of compact chromatin	[2,48]
3. The organization of replication	91% of origin recognition complexes (ORCs) (in interbands)	Only 9% of origin recognition complexes (ORCs) are in the other types of chromatin states	[2] and Figure 5C, this paper
	Early replication	Late replication	
4. Structure of genes in chromosomes			
a) Promoters	Broad promoters	Peaked promoters	[85]
	Motifs revealed: MIBP, BEAF, CRP, ZIPIC	Only a TATA revealed	[64]
b) Intergenic spacers in gene clusters	Median: 706 bp	Median: 3769 bp	Figure 10A
c) Lengths of genes	Median: 2481 bp	Median: 2078 bp	Figure 10A
d) Lengths of exons	Median: 267 bp	Median: 217 bp	Figure 10A
e) Lengths of introns	Median 78 bp	Median: 309 bp	Figure 10B
f) Number of exons	Median 3	Median 4	Figure 10B
g) Number of transcripts	Median 1	Median 2	Figure 10B
5. CHRIZ/CHROMATORmediated transcription	CHRIZ/CHROMATOR plays a role in chromatin decondensation	Unknown proteins	[84]

Hence, this work demonstrated that two groups of genes can be distinguished: housekeeping and developmental ones. These genes differ in terms of many organizational features; in particular, they have different sets of motifs in the promoters, and the structural parts of the genes differ in lengths of the genes, intergenic spaces, the number and size of introns, and the number of transcription starts. In addition, this is according to the degree of chromatin packaging at the locations of these genes, as well as combinations of proteins and histone modifications.

Each gene in both groups is characterized by size and position on the genomic DNA map and can be easily studied individually.

The findings allow further research into these genes. The applied method can be used to study the genomes of organisms having no polytene chromosomes. In particular, it allows one to investigate gene clusters by their co-expression [86,87,88]. The results allow conducting further research into the structures of polytene chromosomes: bands and interbands.

## Figures and Tables

**Figure 1 ijms-25-04068-f001:**
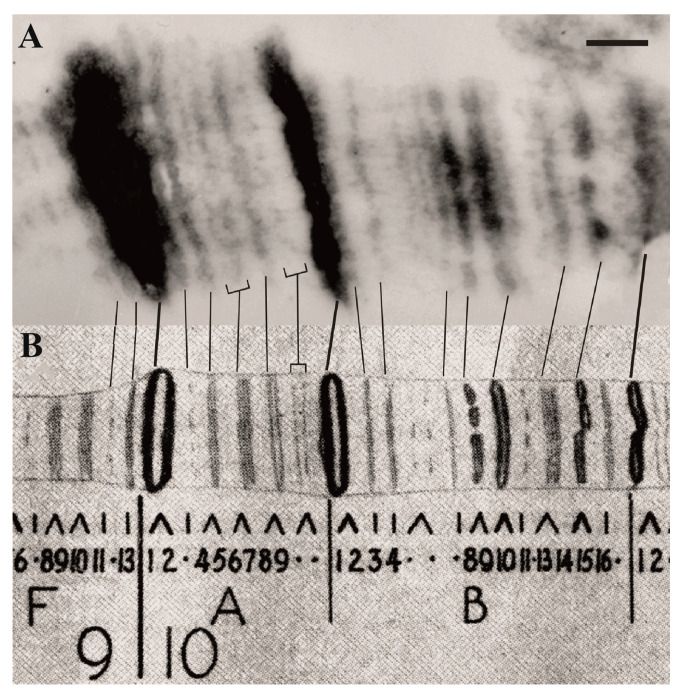
A fragment of the map of the *Drosophila* X chromosome (**A**) according to modern electron microscopy (according to [2]) and (**B**) according to C.B. Bridges [3]. Two types of bands are well distinguishable: black and gray. Scale in “A” corresponds to 1 µm.

**Figure 2 ijms-25-04068-f002:**
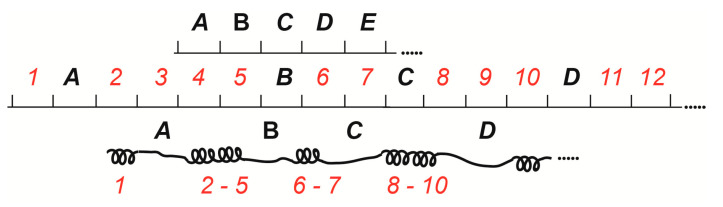
How polytene chromosomes may have become banded. A pre-eukaryotic chromosome shown as a horizontal fiber possesses only housekeeping genes (A–E), whose activity drives the basic metabolic processes in an undifferentiated cell. Genes (1–12) that are active only in a differentiated cell came to the scene later in evolution. Inactivation of the later genes in most cell types; the formation of bands 1, 2–5, 6–7, 8–10, 11–12, and interbands (A–E) (according to [5]).

**Figure 3 ijms-25-04068-f003:**
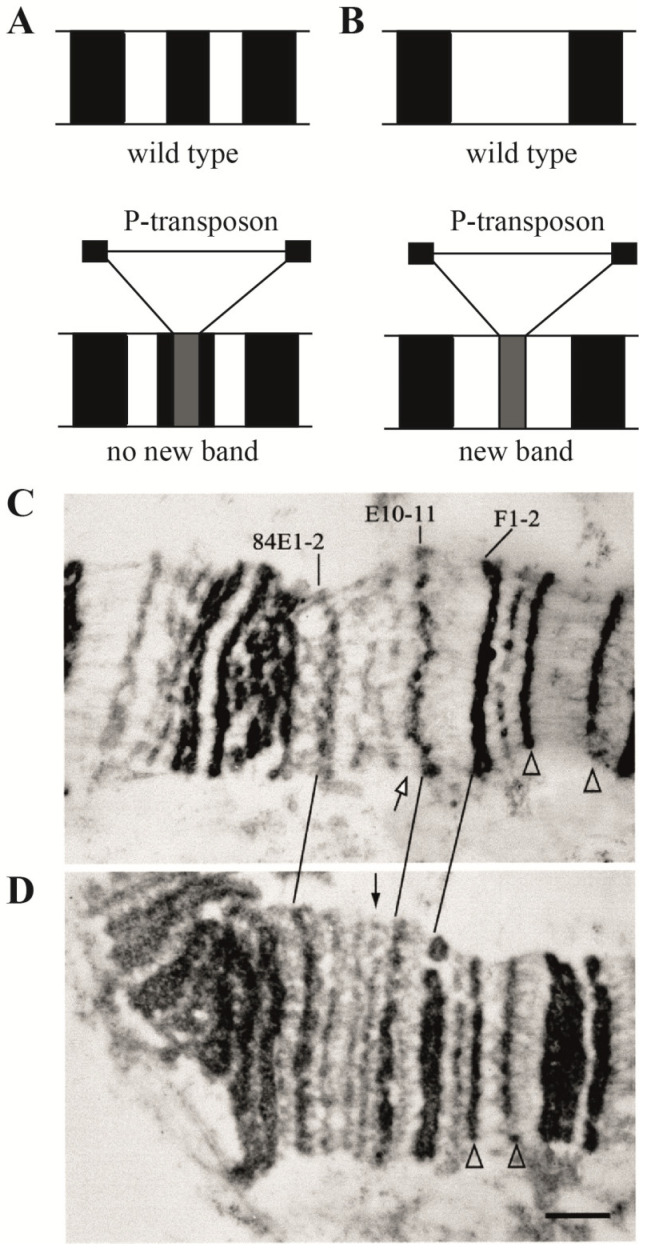
Insertion of the P-transposon in a polytene chromosome interband, principal scheme (**A**,**B**) and electron microscopy (EM) (**C**,**D**) (according to [13,14]). The presence of a transposon insertion was monitored using FISH). Transgenic material forms a novel band (black arrow in (**D**)), which is absent in the wild-type chromosomes (white arrow in (**C**)); some marker bands are shown by arrowheads in (**C**,**D**). The bar corresponds to 1 µm.

**Figure 4 ijms-25-04068-f004:**
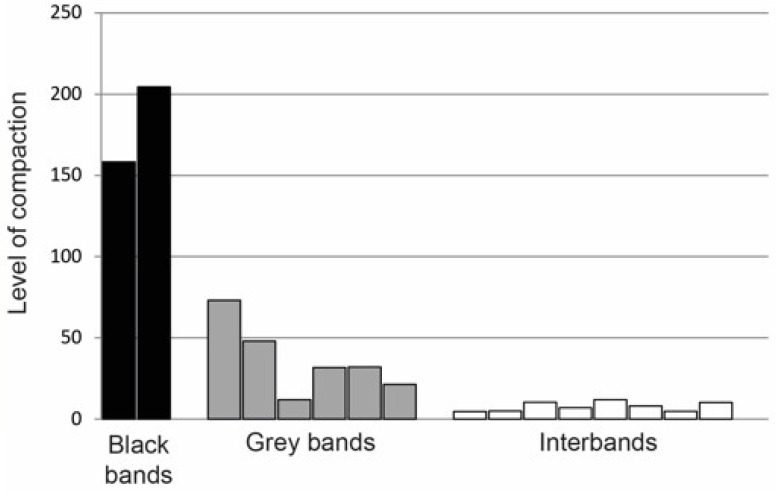
Packing densities of chromosomal material in three types of chromosomal structures: interbands, black, and gray bands in the X chromosome region 101-2-10B1-2 in *Drosophila*. By measuring lengths of the black and gray bands and interbands along the polytene X chromosomes on 50 electron-microscope sections in region 10A-B and the lengths of DNA in these structures, it was found that the packing density of DNA shows a 160–220-fold variation in two black bands, a 20–75-fold variation in gray bands, and a 5–10-fold variation in the interbands (according to [31,32]).

**Figure 5 ijms-25-04068-f005:**
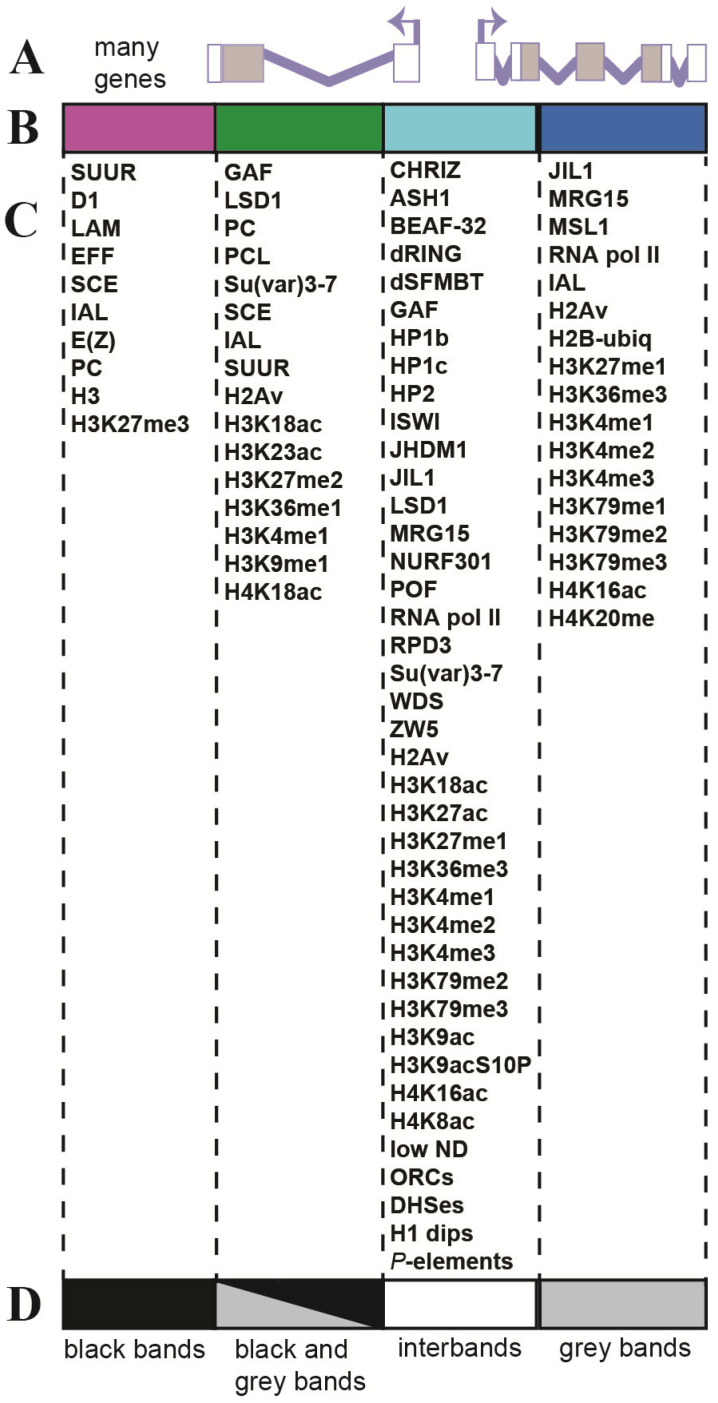
Proteins, histone modifications, and genome elements that stably present on four chromatin domains in S2, Kc167, BG3, and Cl.8 cell lines and in polytene chromosomes of *Drosophila melanogaster*. (**A**)—Gene structure—exons and introns; (**B**)—four states of chromatin domains, left to right: ruby, malachite, aquamarine, and lazurite; (**C**)—proteins and genome elements; (**D**)—types of chromosome structures (gray and black bands, interbands) (according to [2,47,48,49]).

**Figure 6 ijms-25-04068-f006:**
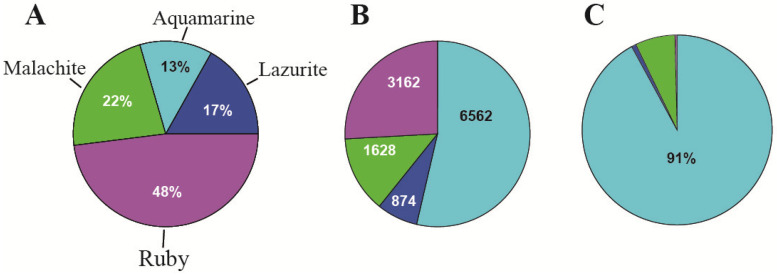
(**A**) Abundance of the four chromatin states in *Drosophila* cells; (**B**) the number of gene promoters in each of the four states; and (**C**) the abundance of ORCs in whole genome of the Kc cells (Malachite, Lazurite, Ruby chromatin state comprises 9%) (5158 ORCs analyzed) (after [2] and all references therein).

**Figure 7 ijms-25-04068-f007:**
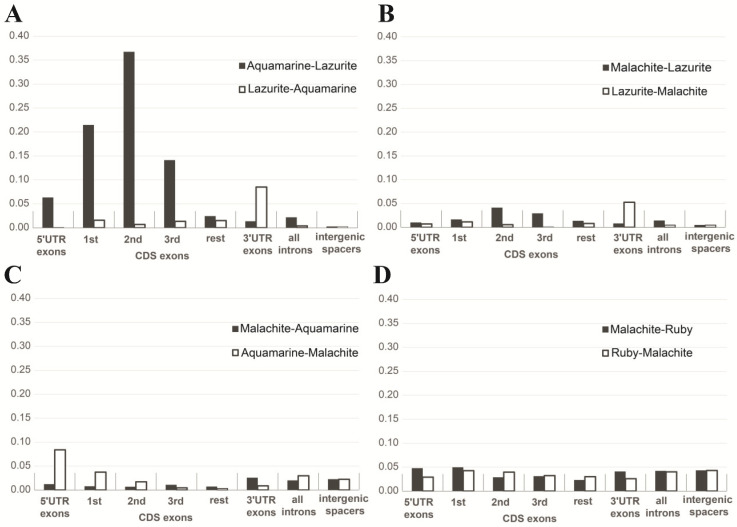
Four chromatin states (black and white bars) in eight possible arrangements relative to each other. Only the aquamarine specifically occurs immediately upstream of the lazurite (**A**), while any other pair is a random event (**B**–**D**) (after [47,48]).

**Figure 8 ijms-25-04068-f008:**
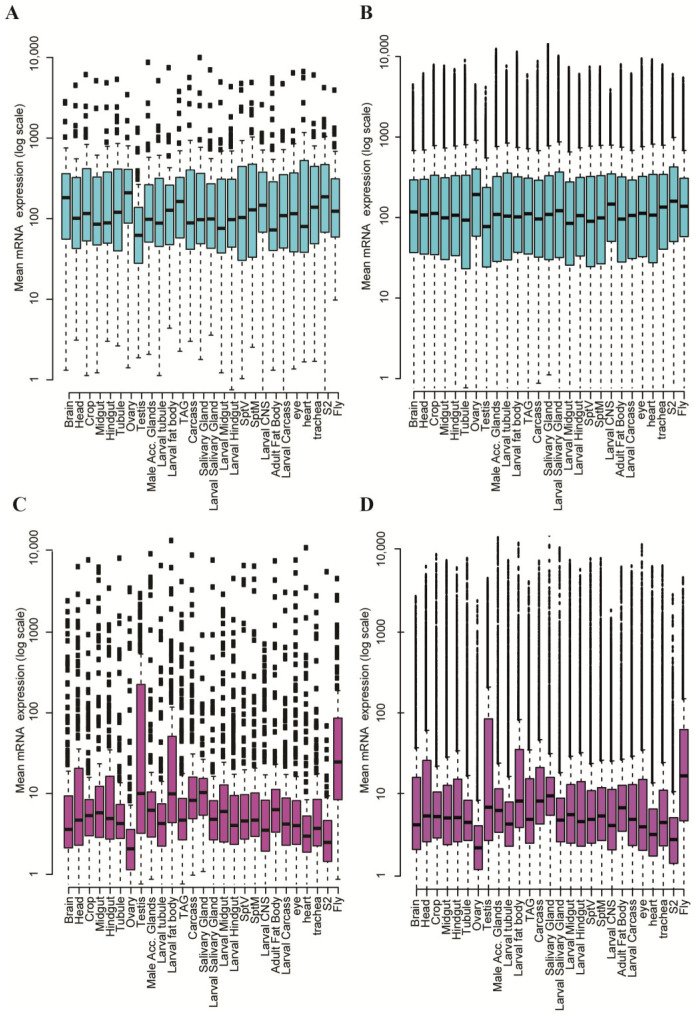
Comparison of the expression levels of the genes located in two types of chromatin: 33 interbands (**A**), full genomic aquamarine fragments (**B**), 33 black bands (**C**), and full genomic ruby (**D**) fragments (from [2,50]). *X* axis: organs and cell types as sources of transcripts; *Y* axis: the log_10_-transformed count of transcript molecules in the cell specimen. Median values (black horizontal lines within the boxes) are shown. Aquamarine and ruby boxes span 50% of the sample size; the dotted lines represent 90% of the sample size; the blank squares designate non-significant values (outliers). A box-and-whisker plot was used (after [2,50,51]).

**Figure 9 ijms-25-04068-f009:**
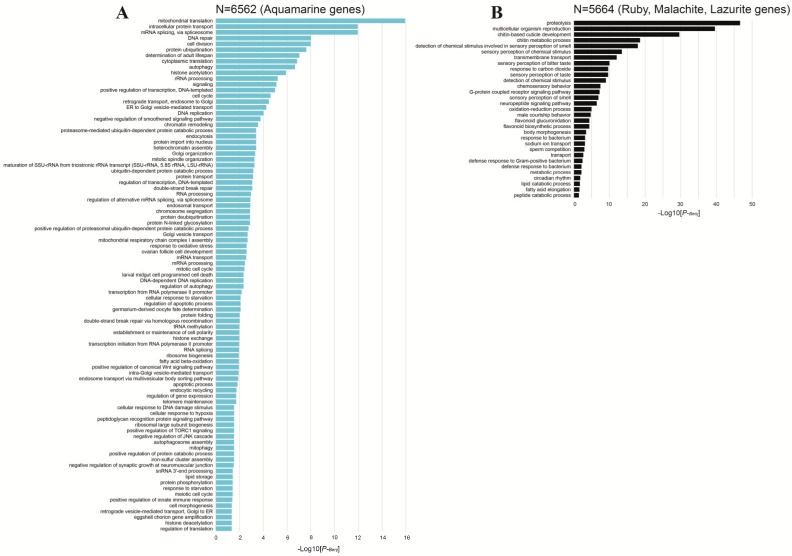
Statistically significant term values for the genes whose promoter regions are located in aquamarine chromatin are shown in blue (**A**). Statistically significant terms for the genes whose promoter regions are located in ruby, malachite, and lazurite chromatins are shown in black (**B**).

**Figure 11 ijms-25-04068-f011:**
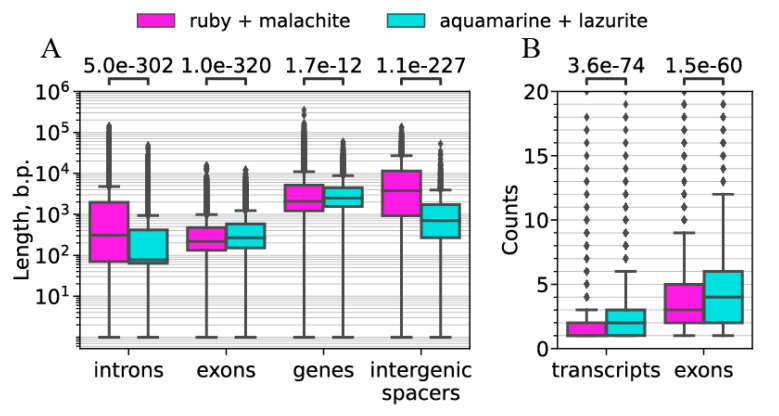
Whole-genome analysis of the structure and chromosomal locations of housekeeping and developmental genes. (**A**) Quartile Q1, Q2, and Q3 distribution of the lengths of the housekeeping and developmental genes, their introns and exons, as well as intergenic spacers. *X* axis: the lengths of genes, introns, exons, and intergenic intervals in the gene clusters. Above the *X* axis: probabilities of similarity (*p*-value). *Y* axis: the number of genes of different lengths (in base pairs). (**B**) Quartile Q1, Q2, and Q3 distribution of the numbers of transcripts and exons in the housekeeping and developmental genes. *X* axis: the transcripts and exons of the genes in the clusters. Above the *X* axis: probabilities of similarity (*p*-value). *Y* axis: the number of transcripts or exons in transcription. For both panes: the error bar values below (Q1) and above (Q3) are the minimum and maximum values, respectively. All the values that fall outside the error bars are outliers (diamonds).

## Data Availability

Not applicable.

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
