# Peer review of "Developmental and Housekeeping Genes: Two Types of Genetic Organization in the Drosophila Genome"

_ijms, 2024, doi:10.3390/ijms25074068_

Round 1
Reviewer 1 Report
Comments and Suggestions for Authors
The presented work provides a detailed description of the organization of Drosophila polytene chromosomes. The authors developed a method for precise localization of genes on polytene chromosomes. This made it possible to describe three types of structure of Drosophila polytene chromosomes. The black bands have been shown to be organized by densely packed inactive developmental genes. The interbands represent gene promoters of housekeeping genes. Gray bars correspond to housekeeping genes. A detailed characterization of housekeeping and developmental genes and their promoters was also carried out. This work is of interest to researchers studying transcription regulation in higher eukaryotes and provides a detailed description of the functional structure of polytene chromosomes, which are one of the best models for studying expression regulation. Below are recommendations for authors.
1) Figure 4 (111-112): the meaning of the sentence is not entirely clear: “If the transposon inserts in a dense band, the band will be visually unaffected, because the transposon's condensed material and the band share the same shade”.
2) Page 4:114-116: It would be better to indicate mentioned “additional approaches” and provide links to relevant manuscripts.
3) Figure 5. “(C) the number of ORCs” – The figure 5 shows the percentage of ORC landing sites in different chromosome regions.
4) Page:4 (140) The number of transcripts in aquamarine chromatin is 22-27 times higher than the number of transcripts
5) Figure 6: The names of organs and tissues in the Figure 6 are difficult to read.
6) Page 7 (172-173). It would be better to indicate which MSL proteins (MSL1, MSL3, MOF?) were identified in gray bands in [21]
7) Table 2: It would be better to rename Table 2 to “Comparison of the properties of developmental and housekeeping genes on Drosophila polytene chromosomes”.
8) Table 2 (2 row): “Maximum where housekeeping genes are located”. May be better: Maximum where housekeeping promoters are located.
9) Table 2 (page 9). I suggest to remove from the Table: “Intergenic spacers in the clusters of developmental genes are five times longer than those between housekeeping genes” and “Genes: 1.2 times longer” etc.
10) Table 2 (page 9): It would be better to remove “5 Proteins in chromatin with clusters of housekeeping and developmental genes” and “6 Transcription processes” from the Table.
11) Figure 8. The title of figure 8 is incomplete. What does the x axis represent?
12) 10 (220) – “contained aquamarine-only material” - contained aquamarine-only chromatin organization?
13) Figure 9. It would be important to show motifs for all mentioned proteins.
According to Flybase, Kni (knirps) expression is not ubiquitous and is only quite high in embryos. Kni primarily acts as a transcriptional repressor associated with negative regulation of enhancers of developmental genes. What role might the Kni protein play in regulating the expression of housekeeping genomes whose expression is ubiquitous?
14) The authors should select the name of protein: CHRIZ/CHRO or CHRIZ/CHROMO (Chromator)
Author Response
Open Review
(x) I would not like to sign my review report
( ) I would like to sign my review report
Quality of English Language
(x) I am not qualified to assess the quality of English in this paper
( ) English very difficult to understand/incomprehensible
( ) Extensive editing of English language required
( ) Moderate editing of English language required
( ) Minor editing of English language required
( ) English language fine. No issues detected
|
Yes |
Can be improved |
Must be improved |
Not applicable |
|
|
Does the introduction provide sufficient background and include all relevant references? |
( ) |
(x) |
( ) |
( ) |
|
Are all the cited references relevant to the research? |
(x) |
( ) |
( ) |
( ) |
|
Is the research design appropriate? |
(x) |
( ) |
( ) |
( ) |
|
Are the methods adequately described? |
(x) |
( ) |
( ) |
( ) |
|
Are the results clearly presented? |
(x) |
( ) |
( ) |
( ) |
|
Are the conclusions supported by the results? |
(x) |
( ) |
( ) |
( ) |
Comments and Suggestions for Authors
The presented work provides a detailed description of the organization of Drosophila polytene chromosomes. The authors developed a method for precise localization of genes on polytene chromosomes. This made it possible to describe three types of structure of Drosophila polytene chromosomes. The black bands have been shown to be organized by densely packed inactive developmental genes. The interbands represent gene promoters of housekeeping genes. Gray bars correspond to housekeeping genes. A detailed characterization of housekeeping and developmental genes and their promoters was also carried out. This work is of interest to researchers studying transcription regulation in higher eukaryotes and provides a detailed description of the functional structure of polytene chromosomes, which are one of the best models for studying expression regulation. Below are recommendations for authors.
First of all we would like to thank the author of this review for his work in critically analyzing our manuscript, where we made numerous corrections. The manuscript is rewritten in many places, which makes it very difficult for you to keep track of line numbers, which have already changed significantly in the new version of the manuscript. To make the work of the reviewer easier, we adhere to the old version of line numbering, and for publication you must use the new version, which is attached.
1) Figure 4 (111-112): the meaning of the sentence is not entirely clear: “If the transposon inserts in a dense band, the band will be visually unaffected, because the transposon's condensed material and the band share the same shade”.
Rewritten (see line number 117-119 in the new version of manuscript)
2) Page 4:114-116: It would be better to indicate mentioned “additional approaches” and provide links to relevant manuscripts.
Rewritten (added several lines in the new version of manuscript: 120-124)
3) Figure 5. “(C) the number of ORCs” – The figure 5 shows the percentage of ORC landing sites in different chromosome regions.
Added to Fig. 5C and in Figure legends
4) Page:4 (140) The number of transcripts in aquamarine chromatin is 22-27 times higher than the number of transcripts
Added word higherin the new version of manuscript
5) Figure 6: The names of organs and tissues in the Figure 6 are difficult to read.
The new version of the figure is done
6) Page 7 (172-173). It would be better to indicate which MSL proteins (MSL1, MSL3, MOF?) were identified in gray bands in [21]
The text has been corrected - MSL1
7) Table 2: It would be better to rename Table 2 to “Comparison of the properties of developmental and housekeeping genes on Drosophila polytene chromosomes”.
Accepted, the text has been corrected
8) Table 2 (2 row): “Maximum where housekeeping genes are located”. May be better: Maximum where housekeeping promoters are located.
Accepted, the text has been corrected
9) Table 2 (page 9). I suggest to remove from the Table: “Intergenic spacers in the clusters of developmental genes are five times longer than those between housekeeping genes” and “Genes: 1.2 times longer” etc.
Accepted, the text has been corrected
10) Table 2 (page 9): It would be better to remove “5 Proteins in chromatin with clusters of housekeeping and developmental genes” and “6 Transcription processes” from the Table.
Accepted and transferred in th text
11) Figure 8. The title of figure 8 is incomplete. What does the x axis represent?
The new version of the figure 8 title of is done
12) 10 (220) – “contained aquamarine-only material” - contained aquamarine-only chromatin organization?
Corrected
13) Figure 9. It would be important to show motifs for all mentioned proteins.
According to Flybase, Kni (knirps) expression is not ubiquitous and is only quite high in embryos. Kni primarily acts as a transcriptional repressor associated with negative regulation of enhancers of developmental genes. What role might the Kni protein play in regulating the expression of housekeeping genomes whose expression is ubiquitous?
The new version of the text with new references written, and new figures E and F are added
14) The authors should select the name of protein: CHRIZ/CHRO or CHRIZ/CHROMO (Chromator)
Yes it is CHRIZ/CHROMATOR
Submission Date
29 December 2023
Date of this review
05 Jan 2024 18:36:28

Reviewer 2 Report
Comments and Suggestions for Authors
This manuscript introduces a novel perspective on gene organization in D. melanogaster. The authors assert that housekeeping genes and developmental genes are arranged in distinct compartments within the genome, as observed in the polytene chromosome structure and banding pattern. The authors attempt to correlate this organization with chromatin structure and the arrangement of bands and interbands.
While the manuscript presents a captivating idea, and Dr. Zhimulev is an esteemed researcher in the field of polytene chromosome structure and organization, there are several issues that currently render the manuscript unsuitable for publication.
Firstly, despite being labeled as a "Brief Report," the manuscript appears more akin to a review of a series of papers primarily produced by the same group.
Furthermore, the writing lacks clarity, and many concepts are poorly presented, causing confusion for the reader. For instance, the connection between the description in paragraph 6 and the manuscript's main topic is unclear. Additionally, some information is missing, hindering the reader's understanding of the main message.
Specific comments are provided below:
l67: This concept could be better articulated. Distinguishing genes based on their expression pattern might be more informative than merely stating "types."
l95: The first draft of the Drosophila melanogaster genome was incomplete as the heterochromatic regions were not fully sequenced and assembled. Only in recent years have we benefited from a comprehensive assembly (doi:10.1038/sdata.2014.45; doi:10.1534/g3.118.200162)
l100-104: This sentence appears unrelated to the initial part of the paragraph; perhaps something is missing.
PlArB (or PlRb): Ensure consistency and introduce it in the main text rather than a figure caption. Reference for this transposon should also be provided.
l108: What is meant by "alien genome"?
l108-109 this behavior should be explained better
l123-127 is this a color code invented by the authors. As described in Kharchenko et al., (doi:10.1038/nature09725) chromatin states are referred as state 1 to state 9. Please explain. If this is a custom designed code explain how this has been determined (chromatin composition, cell type experimental conditions....)
Table 2 Some figures are given as reference. Please cite them as "this work" instead.
The manuscript would benefit from a conclusion paragraph in which the authors highlight how their work make an advancement in the field of genome and chromosomal organization theories.
I hope that the authors can improve the manuscript.
Comments on the Quality of English Language
Some editing is needed
Author Response
Open Review
(x) I would not like to sign my review report
( ) I would like to sign my review report
Quality of English Language
( ) I am not qualified to assess the quality of English in this paper
( ) English very difficult to understand/incomprehensible
( ) Extensive editing of English language required
(x) Moderate editing of English language required
( ) Minor editing of English language required
( ) English language fine. No issues detected
|
Yes |
Can be improved |
Must be improved |
Not applicable |
|
|
Does the introduction provide sufficient background and include all relevant references? |
( ) |
(x) |
( ) |
( ) |
|
Are all the cited references relevant to the research? |
( ) |
( ) |
(x) |
( ) |
|
Is the research design appropriate? |
( ) |
(x) |
( ) |
( ) |
|
Are the methods adequately described? |
( ) |
( ) |
(x) |
( ) |
|
Are the results clearly presented? |
( ) |
( ) |
(x) |
( ) |
|
Are the conclusions supported by the results? |
( ) |
(x) |
( ) |
( ) |
Comments and Suggestions for Authors
This manuscript introduces a novel perspective on gene organization in D. melanogaster. The authors assert that housekeeping genes and developmental genes are arranged in distinct compartments within the genome, as observed in the polytene chromosome structure and banding pattern. The authors attempt to correlate this organization with chromatin structure and the arrangement of bands and interbands.
While the manuscript presents a captivating idea, and Dr. Zhimulev is an esteemed researcher in the field of polytene chromosome structure and organization, there are several issues that currently render the manuscript unsuitable for publication.
First of all we would like to thank the author of this review for his work in critically analyzing our manuscript, where we made numerous corrections. The manuscript is rewritten in many places, which makes it very difficult for you to keep track of line numbers, which have already changed significantly in the new version of the manuscript. To make the work of the reviewer easier, we adhere to the old version of line numbering, and for publication you must use the new version, which is attached.
Firstly, despite being labeled as a "Brief Report," the manuscript appears more akin to a review of a series of papers primarily produced by the same group.
We changed labeling of publication to Brief Review. If it is not acceptable either the Review or Review and Perspectives are also acceptable.
Furthermore, the writing lacks clarity, and many concepts are poorly presented, causing confusion for the reader. For instance, the connection between the description in paragraph 6 and the manuscript's main topic is unclear. Additionally, some information is missing, hindering the reader's understanding of the main message.
The manuscript has been completely revised in many places, rewritten, new figures have been added, Table 2 has been improved, and about a dozen new references have been added.
Specific comments are provided below:
l67: This concept could be better articulated. Distinguishing genes based on their expression pattern might be more informative than merely stating "types."
We could not find how to change it. Below there is complete description of these two types
l95: The first draft of the Drosophila melanogaster genome was incomplete as the heterochromatic regions were not fully sequenced and assembled. Only in recent years have we benefited from a comprehensive assembly (doi:10.1038/sdata.2014.45; doi:10.1534/g3.118.200162)
Accepted, introduced in text
l100-104: This sentence appears unrelated to the initial part of the paragraph; perhaps something is missing.
Sorry, now 10 lines inserted
PlArB (or PlRb): Ensure consistency and introduce it in the main text rather than a figure caption. Reference for this transposon should also be provided.
Reference and short description in the are introduced in the Figure legends of the Fig.4
l108: What is meant by "alien genome"? and l108-109 this behavior should be explained better
Changed in Figure legend of the Figure 4
(Once in another type of cells, the transposon material is inactive and condensed, leading to a compact band-like structure (A and B).
l123-127 is this a color code invented by the authors. As described in Kharchenko et al. (doi:10.1038/nature09725) chromatin states are referred as state 1 to state 9). Please explain. If this is a custom designed code explain how this has been determined (chromatin composition, cell type experimental conditions....)
Sorry, color code is not invention of the authors of this manuscript. For the first time it was used by Filion et al., 2010 (Cell 143, pp. 212-224) and then Kharchenko et al., 2011 (Nature , 471 pp 480-485) . In Figure 1a Kharchenko et al. referred chromatin states are as state 1 to state 9 (and parenthetically give names of colors). Unfortunately colors in Filion and Kharcenko overlap each other, therefore we called the chromatin states by the names of minerals.
As for the second remark of the reviever . Determination ofchromatin composition, cell type experimental conditions and all the others characteristics of our "mineral" model are described in very big separate paper [reference 15 in the manuscript].
Nevertheless we added several sentences to the text to explain the story with the model
Table 2 Some figures are given as reference. Please cite them as "this work" instead.
The Table 2 is changed in several places.
The manuscript would benefit from a conclusion paragraph in which the authors highlight how their work make an advancement in the field of genome and chromosomal organization theories.
I hope that the authors can improve the manuscript.
Comments on the Quality of English Language
Some editing is needed
Submission Date
29 December 2023
Date of this review
10 Jan 2024 17:00:07

Round 2
Reviewer 2 Report
Comments and Suggestions for Authors
In the modified version of the manuscript, the authors have made some corrections. However, the manuscript has not been significantly improved and it has not reached the publication level. The manuscript seems like a patchwork of information making it difficult for the reader to follw and it is challenging to discern the core message of the work is. If the authors' goal is to summarize the results obtained in their previous works then they should find a comprehensive and clear way to do so.
Below are listed some of the issues associated to the manuscript.
Figure 1 is identical to figure 1 in another publication from the same group (doi:10.1002/bies.201100142)
l97 the two references are not properly formatted
l101 the discovery of mobile elements is attributed to B. McClintock, so the references here are not correct.
l104-106 Is this claim a rule? Or it is an observation related to some transposons? I don't believe that all TE insertions are inactivated and all can for a visible band.
l110-111 "...in this transposon inserion site...", "...Drosophila line that carries 110 this transposon..." what transposon are the authors referring to?
l116 the description of PlrB provided in figure 4 caption is partially wrong
l120 In the work cited in this paragraph (Semeshin et al 1989) no mention is made to the utilization of P1lrB. So, I do not understand why this transposon has been introduced in the previous paragraph.
Concerning the color code, as far as I can see from their response the authors have assigned the name of minerals to chromatin states and given that nor Filion nor Kharchenko have used those names, it sounds like this is an author's personal classification. As such, it should be clearly explained the association between this classification and the currently used chromatin states. Reference 15 does not contain this explanation (I checked reference 15 on both versions of the ms but I was not able to find this information). Furthermore, without this clarification it is not possible to explain why lazurite and malachite represent different chromatin states.
l182 "...these antibodies are located in gene bodies..." is that correct?
l238-241 this claim at least needs a reference
l248-250 this sentence need to be clarified
l344-348 this conclusion is obvious and it is not in the context of the black-gray-interbands structure of the polytene chromosomes
Several typos are present in the manuscript, which calls for an extensive language editing
Examples: 104 transpozone; l105 transpozon; l110 inserion; l265 chromatine
Comments on the Quality of English Languagethe manuscript needs a profound revision
Author Response
- x) I would not like to sign my review report
( ) I would like to sign my review report
Quality of English Language
( ) I am not qualified to assess the quality of English in this paper
( ) English very difficult to understand/incomprehensible
(x) Extensive editing of English language required
( ) Moderate editing of English language required
( ) Minor editing of English language required
( ) English language fine. No issues detected
Comments and Suggestions for Authors
In the modified version of the manuscript, the authors have made some corrections. However, the manuscript has not been significantly improved and it has not reached the publication level. The manuscript seems like a patchwork of information making it difficult for the reader to follow and it is challenging to discern the core message of the work is. If the authors' goal is to summarize the results obtained in their previous works then they should find a comprehensive and clear way to do so.
In this new version, part of the manuscript has been revised: ambiguities in the presentation have been removed, Table 1 has been removed and replaced with a more compact drawing, and T. Paynter’s drawing has been removed. The citation of various works by other authors, which were omitted in the first version, has been significantly increased, in order to convey the main idea as simply as possible - the most accurate and original identification of two groups of genes in the Drosophila genome, a description of the features and differences of each of them. We have tried to make the presentation more understandable.
The purpose of the work, in our opinion, is clearly indicated already in the title and runs through the entire article - to describe the discovery and rationale for the existence of two types of genes - development and housekeeping. In the manuscript this is the last part and it is the main one. On this topic, not a single reviewer has a single comment about this material, and this is precisely the basis of the article. Although many scientists have discussed the existence of these groups of genes, the work we have done and presented in the manuscript is a priority, since now we know how many genes (with currently high accuracy) in the Drosophila genome belong to each of the groups, in what coordinates on the DNA map each of the thousands of genes is located, in what chromosomal structures it lies in the genome, how the genes are located in clusters (lying in black disks), what are the sizes and organization of gene structures (motifs in gene promoters, lengths of genes, introns, exons, number of transcripts from each gene, sizes of intergenic spaces). What are the proteins that serve to organize transcription or maintain genes in an inactive state in these groups of genes, and how genes of both types are involved in organizing replication. On all these points the housekeeping and developmental genes are quite different. These are the main conclusions, completely new and unexpected. There was not a single comment on them either from the first reviewer or in the two reviews of the second. As for the first part of the manuscript, we showed how we approached the creation of a bioinformatics model of the organization of the Drosophila genome, which allows us to identify these groups of genes. This work took us many years, was carried out using completely different techniques in cytology and electron microscopy, as well as in molecular biology. Therefore, it is not surprising that it is not always completely clear to the same scientist.
Below are listed some of the issues associated to the manuscript.
Figure 1 is identical to figure 1 in another publication from the same group (doi:10.1002/bies.201100142)
Figure 1 is not ours. It is taken from an article by the discoverer of polytene chromosomes - a remarkable American zoologist from the University of Texas - Theophilus Shickel Painter, (1889 - 1969), who described the polytene chromosomes of Drosophila in 1934, i.e. this is not self-citation. Since not many scientists quote this remarkable scientist, and have rarely even heard of him, we believed that we could cite the world's first drawing of polytene chromosomes from his publication along with a link to his article as many times as we think is necessary. However, if this is a "problem", a reason for comment and, in fact, a reason for rejecting the article, we have removed the figure from the manuscript.
l97 the two references are not properly formatted
Changed in manuscript.
l101 the discovery of mobile elements is attributed to B. McClintock, so the references here are not correct.
The manuscript states that we are talking about Drosophila and the possibility of working with its DNA. Nevertheless, one additional phrase has been added in new version.
l104-106 Is this claim a rule? Or it is an observation related to some transposons? I don't believe that all TE insertions are inactivated and all can for a visible band.
In all cases studied.
l110-111 "...in this transposon inserion site...", "...Drosophila line that carries 110 this transposon..." what transposon are the authors referring to?
We added a new sentences indicated that there were control experiments with FISH and showed on EM (new Fig. 3) how th new band looks like. We have given a reference. All these papers were published long ago.
l116 the description of PlrB provided in figure 4 caption is partially wrong
Thanks. We changed this.
l120 In the work cited in this paragraph (Semeshin et al 1989) no mention is made to the utilization of P1lrB. So, I do not understand why this transposon has been introduced in the previous paragraph.
We have given detailed version of description with references .
Concerning the color code, as far as I can see from their response the authors have assigned the name of minerals to chromatin states and given that nor Filion nor Kharchenko have used those names, it sounds like this is an author's personal classification. As such, it should be clearly explained the association between this classification and the currently used chromatin states. Reference 15 does not contain this explanation (I checked reference 15 on both versions of the ms but I was not able to find this information). Furthermore, without this clarification it is not possible to explain why lazurite and malachite represent different chromatin states.
We have given a new piece of text
l182 "...these antibodies are located in gene bodies..." is that correct?
Changed
l248-250 this sentence need to be clarified
Changed
l344-348 this conclusion is obvious and it is not in the context of the black-gray-interbands structure of the polytene chromosomes
Removed
Several typos are present in the manuscript, which calls for an extensive language editing
Examples: 104 transpozone; l105 transpozon; l110 inserion; l265 chromatine
Changed
Comments on the Quality of English Language
the manuscript needs a profound revision
The manuscript was looked through by professional interpreter now.
Submission Date
29 December 2023
Date of this review
29 Jan 2024 20:12:16

Round 3
Reviewer 2 Report
Comments and Suggestions for Authors
I thank the Authors for the clarification provided in the rebuttal letter.
There are still issues associated to the manuscript
- Chromatin state color code. Can a clear chromatin composition (in terms of histone modifications) be defined for each state? If yes please insert a description (in form of text or table). If not explain why (not experimentally tested?)
106-107 some text in russian should be either translated or removed
- In my previous report I pointed out that the description of the PlArB vector was. inaccurate. In the current manuscript the authors have removed the (wrong) description from the caption to figure 4 and relocated it (again wrong) in the text (l107). Indeed, as far as I can see from the original paper that describes the PlArb vector it does not carry the Bar gene. Please revise.
- As stated in my first revision report, the manuscript would benefit from a conclusion paragraph in which the authors highlight how their work make an advancement in the field of genome and chromosomal organization theories. Please also highlight limitations of this work. Some of the information that the authors provided in the rebuttal letter can be used to conclude the manuscript.
- several typos are still present throughout the manuscript
Comments on the Quality of English Language
Manuscript needs some text editing
Author Response
Open Review
(x) I would not like to sign my review report
( ) I would like to sign my review report
Quality of English Language
( ) I am not qualified to assess the quality of English in this paper
( ) English very difficult to understand/incomprehensible
( ) Extensive editing of English language required
(x) Moderate editing of English language required
( ) Minor editing of English language required
( ) English language fine. No issues detected
Comments and Suggestions for Authors
I thank the Authors for the clarification provided in the rebuttal letter.
There are still issues associated to the manuscript
- Chromatin state color code. Can a clear chromatin composition (in terms of histone modifications) be defined for each state? If yes please insert a description (in form of text or table). If not explain why (not experimentally tested?)
On the one hand, they constantly demand to reduce self-citation, on the other hand, they demand more and more details and other data. How to do this when you need to reduce the number of your articles. However, the only option remains to provide links to brief mentions in already cited articles, which is what we did (see Table 1 and links in appropriate places in the text).
106-107 some text in russian should be either translated or removed
Corrected
- In my previous report I pointed out that the description of the PlArB vector was. inaccurate. In the current manuscript the authors have removed the (wrong) description from the caption to figure 4 and relocated it (again wrong) in the text (l107). Indeed, as far as I can see from the original paper that describes the PlArb vector it does not carry the Bar gene. Please revise.
There are different spellings for the name of this transposon: the authors themselves used two in their work (Wilson et al., 1989; Gene Developm, 3, pp. 1301-1313): P[LArB] and P-lArB (see abstract, fig. .1 and figure legend to Fig. 1), and the reviewer himself uses another method - the PlArB vector (see above). To minimize the problems associated with this completely minor issue, we have minimized the mention of the name of this transposon in the manuscript.
As stated in my first revision report, the manuscript would benefit from a conclusion paragraph in which the authors highlight how their work make an advancement in the field of genome and chromosomal organization theories. Please also highlight limitations of this work. Some of the information that the authors provided in the rebuttal letter can be used to conclude the manuscript.
Accepted and made at the very end of the manuscript
- several typos are still present throughout the manuscript
Made
Comments on the Quality of English Language
Manuscript needs some text editing
Submission Date
29 December 2023
Date of this review
29 Feb 2024 11:38:51
